# Complex Regulation of Gamma-Hemolysin Expression Impacts *Staphylococcus aureus* Virulence

Mariane Pivard,[a]* Isabelle Caldelari,[b] Virginie Brun,[c,d] Delphine Croisier,[e] Michel Jaquinod,[c] Nelson Anzala,[e] Benoît Gilquin,[c,d] Chloé Teixeira,[a] Yvonne Benito,[f,g] Florence Couzon,[a] Pascale Romby,[b] Karen Moreau,[a] François Vandenesch[a,f,g]

[a]CIRI, Centre International de Recherche en Infectiologie, Université de Lyon, Inserm U1111, Université Claude Bernard Lyon 1, CNRS UMR5308, ENS de Lyon, Lyon, France
[b]Architecture et Réactivité de l'ARN, Université de Strasbourg, CNRS, IBMC, Strasbourg, France
[c]Université Grenoble Alpes, Inserm, CEA, UMR BioSanté U1292, CNRS, CEA, Grenoble, France
[d]Université Grenoble Alpes, CEA, LETI, Clinatec, Grenoble, France
[e]Vivexia, Dijon, France
[f]Institut des Agents Infectieux, Hospices Civils de Lyon, Lyon, France
[g]Centre National de Référence des Staphylocoques, Hospices Civils de Lyon, Lyon, France

Pascale Romby, Karen Moreau, and François Vandenesch contributed equally to this work.

**ABSTRACT** *Staphylococcus aureus* gamma-hemolysin CB (HlgCB) is a core-genome-encoded pore-forming toxin that targets the C5a receptor, similar to the phage-encoded Panton-Valentine leucocidin (PVL). Absolute quantification by mass spectrometry of HlgCB in 39 community-acquired pneumonia (CAP) isolates showed considerable variations in the HlgC and HlgB yields between isolates. Moreover, although HlgC and HlgB are encoded on a single operon, their levels were dissociated in 10% of the clinical strains studied. To decipher the molecular basis for the variation in *hlgCB* expression and protein production among strains, different regulation levels were analyzed in representative clinical isolates and reference strains. Both the HlgCB level and the HlgC/HlgB ratio were found to depend on *hlgC* promoter activity and mRNA processing and translation. Strikingly, only one single nucleotide polymorphism (SNP) in the 5′ untranslated region (UTR) of *hlgCB* mRNA strongly impaired *hlgC* translation in the USA300 strain, leading to a strong decrease in the level of HlgC but not in HlgB. Finally, we found that high levels of HlgCB synthesis led to mortality in a rabbit model of pneumonia, correlated with the implication of the role of HlgCB in severe *S. aureus* CAP. Taken together, this work illustrates the complexity of virulence factor expression in clinical strains and demonstrates a butterfly effect where subtle genomic variations have a major impact on phenotype and virulence.

**IMPORTANCE** *S. aureus* virulence in pneumonia results in its ability to produce several virulence factors, including the leucocidin PVL. Here, we demonstrate that HlgCB, another leucocidin, which targets the same receptors as PVL, highly contributes to *S. aureus* virulence in *pvl*-negative strains. In addition, considerable variations in HlgCB quantities are observed among clinical isolates from patients with CAP. Biomolecular analyses have revealed that a few SNPs in the promoter sequences and only one SNP in the 5′ UTR of *hlgCB* mRNA induce the differential expression of *hlgCB*, drastically impacting *hlgC* mRNA translation. This work illustrates the subtlety of regulatory mechanisms in bacteria, especially the sometimes major effects on phenotypes of single nucleotide variation in noncoding regions.

**KEYWORDS** *Staphylococcus aureus*, gamma-hemolysin CB, toxin regulation, mRNA translation, pneumonia, rabbit model

Address correspondence to François Vandenesch, francois.vandenesch@univ-lyon1.fr.

*Present address: Mariane Pivard, Department of Infectious Diseases and Hospital Epidemiology, University Hospital Zurich, University of Zurich, Zurich, Switzerland.

The authors declare no conflict of interest.

The Gram-positive bacterium *Staphylococcus aureus* can provoke various severe diseases in humans, such as bacteremia, endocarditis, and pneumonia. Its capacity to develop a wide range of infections is due to its ability to produce a large diversity of virulence

factors, including immune evasion molecules, adhesins, and toxins (1, 2). *S. aureus* accounts for approximately 25% of nosocomial pneumonia cases (3) and can be responsible for severe community-acquired pneumonia (CAP) with high mortality rates (4–7). In the latter context, the bicomponent pore-forming toxin (PFT) Panton-Valentine leucocidin (LukSF-PV) (also designated PVL), which targets the C5a and C5L2 receptors on neutrophils, monocytes, and macrophages and the CD45 receptor on leukocytes (8, 9), has been associated with severity independent of other virulence determinants and methicillin resistance (4). Nevertheless, although PVL is present in almost 50% of strains isolated from patients with severe CAP (4), the remaining cases are caused by PVL-negative strains and are still associated with a 30% mortality rate (4), suggesting that other virulence factors are involved. An obvious candidate is gamma-hemolysin CB (HlgCB) (encoded in the *hlgCB* operon), which is present in almost all strains (10) and targets the C5a receptors on myeloid cells, as does PVL (11, 12). Of note, HlgB also associates with HlgA (encoded by *hlgA* immediately upstream of *hlgCB*) to comprise HlgAB, a toxin targeting CXCR1, CXCR2, and CCR1 (11, 12). Moreover, a recent semiquantitative proteomic analysis of exotoxins applied to 136 strains isolated from cases of severe CAP showed that a high level of HlgB production in PVL-negative strains was independently associated with hemoptysis (13), a major severity parameter in CAP (4). In addition, that study showed that the HlgB and HlgC levels were not correlated, in contrast to other operon-encoded virulence factors (13), and a weak negative correlation between LukS-PV and HlgC was observed.

In the present study, we have investigated the molecular mechanisms leading to HlgC and HlgB expression variations among clinical isolates and assessed whether HlgCB is a major determinant of pneumonia severity in a rabbit model (14). Besides identifying a new translatable *hlgB* mRNA as a maturation product of *hlgCB*, the data show that the variation in HlgC and HlgB production results from at least three parameters: (i) promoter strength; (ii) translation efficiency, which is dependent on a single nucleotide polymorphism (SNP) in the 5′ untranslated region (UTR) of *hlgC* mRNA; and (iii) *hlgCB* mRNA stability. Furthermore, high levels of expression of HlgCB induced high mortality rates in the rabbit model, which could also contribute to the severity of *S. aureus* CAP in humans.

## RESULTS

**High levels of variation in HlgCB production among clinical strains: two distinct profiles.** To validate the results of a previous semiquantitative proteomic analysis of exotoxins (15), the two hemolysins HlgC and HlgB were quantified using isotope-dilution-targeted proteomics in a subset of 39 clinical isolates representative of the genomic diversity of the cohort (maintained proportions of clonal complexes and *pvl*-positive strains). The USA300 SF8300 (USA300) and HT20020209-LUG1799 (ST80) strains were used as control strains because their exoprotein profiles and virulence in various animal models are well defined (16, 17). Absolute quantification showed considerable HlgC and HlgB variations among isolates with two distinct profiles. In four strains, including USA300, the level of the HlgB protein was 6 to 14 times higher than that of HlgC, which was very low. In most of the other strains, including the ST80 strain and the PVL-negative PEN426 (PEN) strain, isolated from a deceased patient (4), similar levels of HlgC and HlgB were observed, and in the majority of the cases, the levels of HlgB together with HlgC were higher than those of the isolates of profile 2 (Fig. 1).

**High levels of HlgCB impact severity in a rabbit model of pneumonia.** To experimentally analyze if high levels of HlgCB may influence the virulence of *S. aureus*, the PEN strain (high-level producer of HlgCB and PVL negative) was tested in a rabbit model of pneumonia. Its virulence was compared to those of USA300 and ST80, both highly lethal PVL-positive methicillin-resistant *S. aureus* (MRSA) strains (16, 17). All infected rabbits died within 48 h (Fig. 2A), with high lung bacterial densities and high ratios of lung weight to body weight (LW/BW) (Fig. 2B and C). To determine whether the virulence of the PEN strain was caused by its high level of production of HlgCB, a premature stop codon was inserted into the *hlgC* gene (PEN<Q63X) to prevent the production of HlgC and HlgCB. Proteomics indicated that HlgB was still produced in

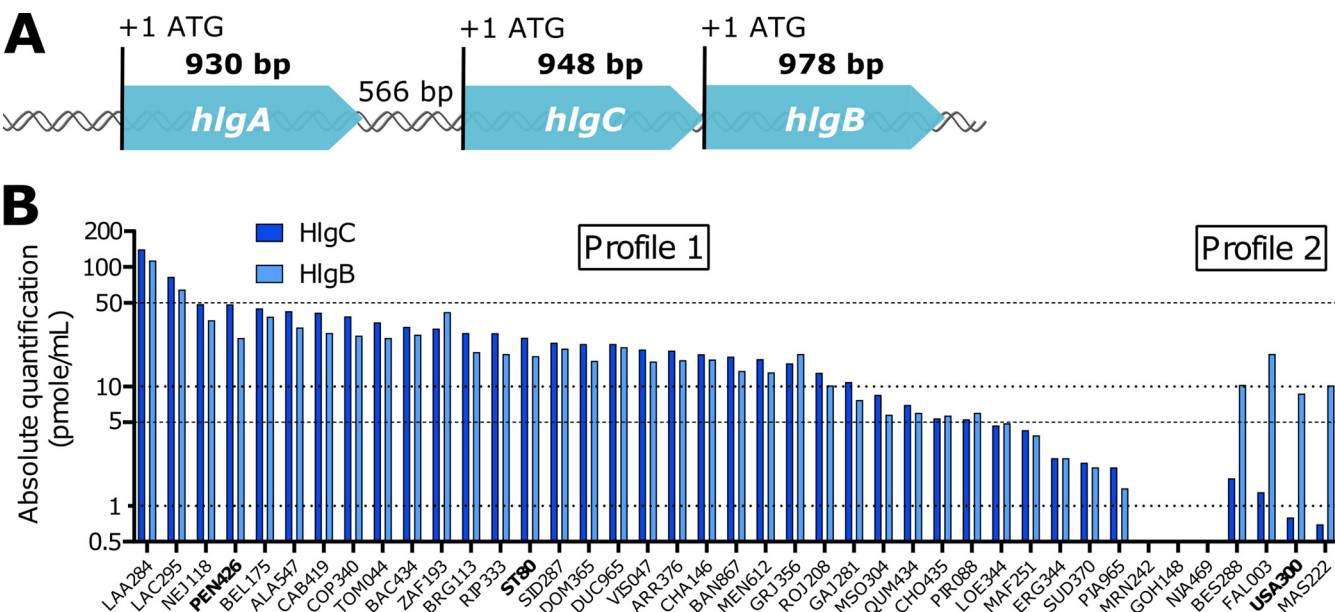

**FIG 1** The gamma-hemolysin genomic locus: two distinct profiles and high levels of variation in HlgCB secretion among clinical isolates. (A) Schematic representation of the gamma-hemolysin genomic locus. The sizes of the coding and intergenic regions are expressed in base pairs. (B) HlgC and HlgB were quantified using isotope-dilution-targeted proteomics on *S. aureus* culture supernatants. ST80, USA300, and 39 clinical isolates from patients with severe CAP were cultured in CCY medium for 8 h, and the supernatants were collected and analyzed by targeted mass spectrometry. Quantifications are represented on a $\log_{10}$ scale in picomoles per milliliter. Two distinct profiles are observed: similar levels of HlgC and HlgB on the left side and higher levels of HlgB than those of HlgC on the right side. For three strains, the levels of HlgC and HlgB were below the detection limit. The three strains explored in detail in this work, USA300, ST80, and PEN426 (referred to as PEN here) are in boldface type.

PEN<Q63X albeit at a lower level than that in the wild-type PEN (PEN WT) strain (see Table S1 in the supplemental material). Rabbits infected with PEN<Q63X showed significantly greater survival (Fig. 2A) and lower bacterial densities and LW/BW ratios (Fig. 2B and C) than rabbits infected with PEN WT. In addition, no inflammatory response was observed in rabbits infected by PEN<Q63X, whereas similar levels of interleukin-1$\beta$ (IL-1$\beta$) and IL-8 were quantified in rabbits infected by the USA300, ST80, and PEN WT strains (Fig. 2D). To determine how this toxin participates in severity in the case of strains also producing PVL, specific gene deletions were performed in the ST80 strain either individually in the genes encoding PVL, HlgC, and Hla or simultaneously at all genes encoding toxins (triple mutant). The results reveal that in this context, only the triple mutant strain had its virulence considerably attenuated, suggesting the existence of compensation between the different toxins (Fig. S1). These data strongly demonstrate that, at least in PVL-negative strains, high levels of HlgCB are correlated with severe outcomes and mortality in a manner similar to that of PVL-positive strains like USA300 and ST80.

**Analysis of the *hlgCB* operon reveals a processed *hlgB* transcript.** To compare the proteomic profiles of the USA300, ST80, and PEN strains with RNA levels, Northern blot analysis was performed on the three strains using specific probes complementary to *hlgA*, *hlgC*, and *hlgB* mRNAs. The *hlgC* and *hlgB* probes unveiled the expected bicistronic *hlgCB* transcript (Fig. 3A). Significant variation in *hlgCB* expression was observed, with PEN producing more *hlgCB* mRNA than USA300 and ST80. As expected, no correlation between *hlgCB* and *hlgA* mRNA expression levels was found. Indeed, both USA300 and PEN yielded more *hlgA* mRNA than ST80, although ST80 produced more *hlgCB* mRNA than USA300. These data confirm that the bicistronic *hlgCB* mRNA is transcribed under the control of its own promoter. In addition, the *hlgB*-specific probe revealed the presence of a smaller product, the size of which corresponded to *hlgB* alone. Moreover, the *hlgB* transcript level was much higher than that of *hlgCB* in USA300, while the yields of the two transcripts were roughly equivalent in the ST80 and PEN strains (Fig. 3A).

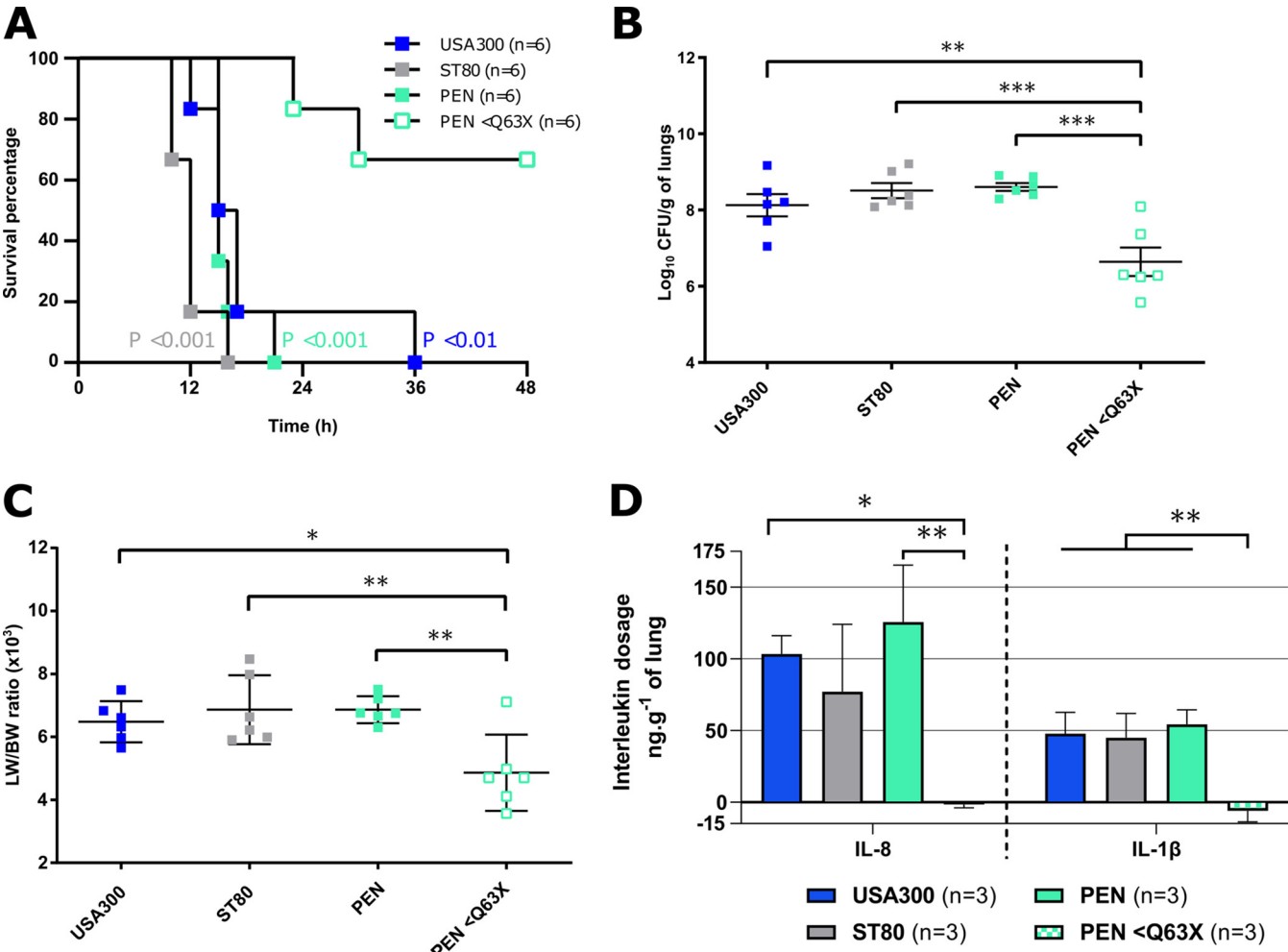

**FIG 2** HlgCB contributes to *S. aureus* virulence during pneumonia. (A) Kaplan-Meier survival curves for rabbits infected by the endotracheal instillation of 9.5 to 9.6 $\log_{10}$ CFU/mL of USA300, ST80, PEN, or PEN with a premature stop codon in *hlg*C (PEN<Q63X) to induce necrotizing pneumonia. A log rank (Mantel-Cox) test was used to compare the rate of mortality for PEN HlgC<Q63X to those for USA300, ST80, and PEN. (B to D) Data on bacterial densities in the lung on a $\log_{10}$ scale (*n* = 6) (B), ratios of lung weight to body weight (LW/BW) ($\times 10^3$) (*n* = 6) (C), and ELISA quantification of IL-8 and IL-1$\beta$ concentrations in lung shred samples reported relative to the total lung mass (*n* = 3) were collected from rabbits infected with the USA300, ST80, PEN, and PEN HlgC<Q63X strains and compared by Tukey's multiple-comparison test. *, $P < 0.05$; **, $P < 0.01$; ***, $P < 0.001$ (nonsignificant comparisons are not shown).

We then investigated whether the *hlgB* transcript was under the control of a specific promoter upstream of *hlgB*. The 932 bp upstream of the *hlgB* start codons of the USA300 and ST80 strains (with the sequence of PEN being strictly identical to that of USA300) were cloned into the pALC1484 plasmid in front of the green fluorescent protein (GFP) gene (*gfp*). GFP signal quantification failed to detect any promoter activity, as demonstrated by the absence of a signal for both the *hlgB*-upstream region tested (pALC-P*hlgB*) and the empty vector (pACL-ø), unlike the positive control (pACL-P16S) containing the 16S-encoding promoter (Fig. 3B and Fig. S2A) and the *hlgC* promoter (P*hlgC*) in both strains (Fig. S2B). The transcriptional start sites of the *hlgB* and *hlgCB* transcripts were then characterized by primer extension in ST80 and USA300 (Fig. S3). The results showed three 5′ ends located at positions −80, −104, and −136 upstream of the *hlgB* start codon and only one stop at position −35 upstream of the *hlgC* start codon, all possible transcriptional start sites (Fig. 3C and Fig. S3). Since bona fide mRNA can be distinguished from processed RNA by the presence of a 5′-triphosphate, the Terminator 5′-phosphate-dependent exonuclease, which preferentially digests RNA carrying a 5′-monophosphate, was used. Northern blot experiments showed the full degradation of the short *hlgB* mRNA in the three strains, while the control bona fide small RNA (sRNA) RsaI was resistant (Fig. S4), reinforcing the involvement of a processing event. Of note, *hlgCB* was also sensitive to enzymatic treatment. We

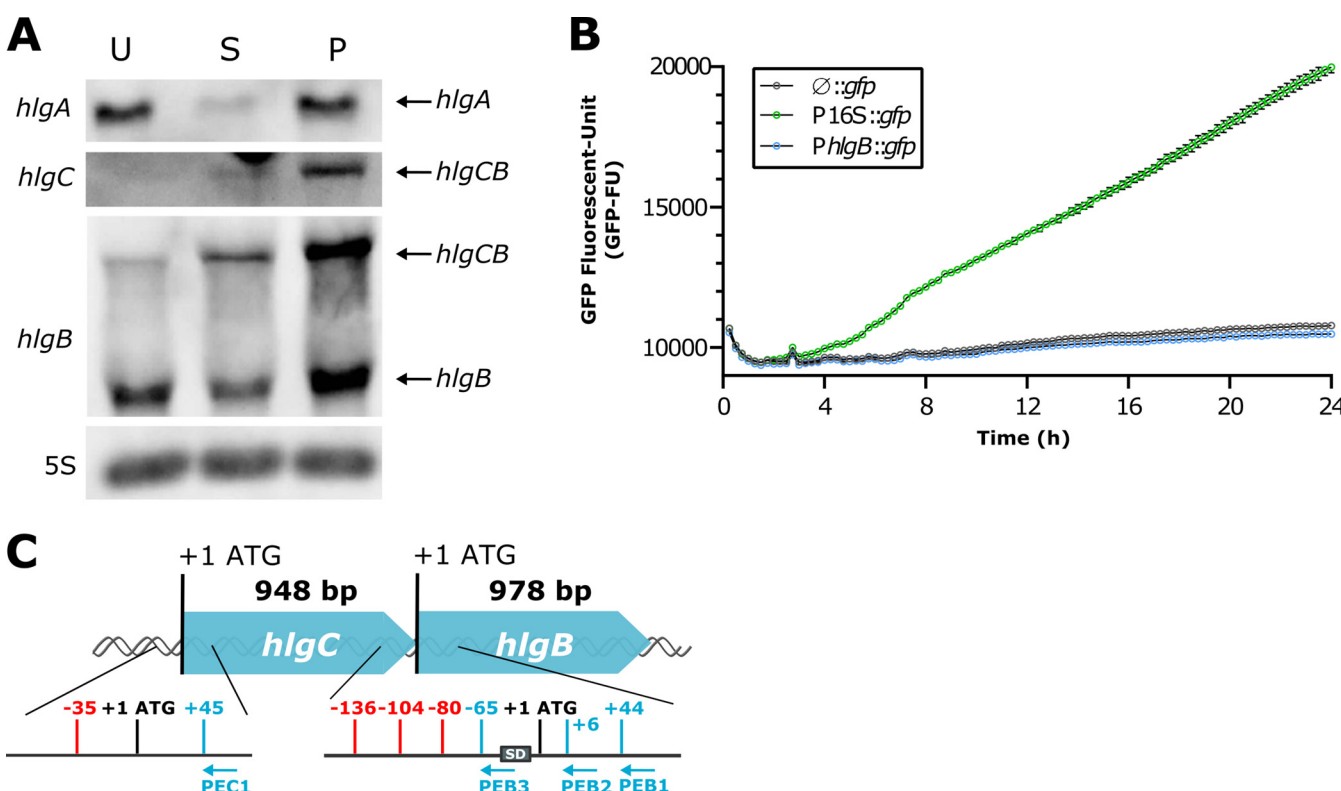

**FIG 3** Different expression levels of *hlgA* and *hlgCB* mRNAs in three clinical strains: an additional *hlgB* transcript. (A) Northern blot analysis of gamma-hemolysin expression in the USA300 (U), ST80 (S), and PEN (P) strains. Total RNA was extracted from cultures grown for 8 h in CCY medium. Probes targeting *hlgA*, *hlgC*, and *hlgB* mRNAs were used and are indicated on the left side. The *hlgB* probe detected two transcripts, *hlgCB* mRNA and *hlgB* mRNA, as mentioned on the right side. 5S rRNA (5S) was used as a loading control. Blots are representative of results from triplicate biological experiments. (B) The existence of an *hlgB*-specific promoter within the *hlgC* gene was tested using transcriptional fusions in the USA300 strain. The 926-bp region upstream of *hlgB*, including the *hlgB* ATG codon, was cloned into the pACL-1484 plasmid in front of the *gfp* gene (P*hlgB*::gfp). The 16S rRNA promoter was used as a positive control (P16S::*gfp*), and the plasmid with no insert was used as a negative control (ø::*gfp*). Promoter activity was measured by GFP signal quantification (GFP fluorescence units) over time, during a 24-h kinetic with a 15-min interval, in CCY medium containing chloramphenicol. Means with standard errors from biological triplicates are represented for pACL-ø, pACL-P16S, and pACL-*hlgB*. (C) Schematic summary of the primer extension results. Primer extension was performed on total RNA from the USA300 and ST80 strains using 5′-radiolabeled primers complementary to the *hlgC* (PEC1) and *hlgB* (PEB1, PEB2, and PEB3) sequences. All identified mRNA start sites are in red, primers are in blue, and the *hlgB* Shine-Dalgarno (SD) sequence is represented by a gray box.

do not exclude that the *S. aureus* RNase J1/J2, which is active on mRNAs carrying 5′-triphosphate, partially cleaved the 5′ end of *hlg*C.

**Promoter activity and posttranscriptional mechanisms impact *hlgCB* expression.** We then assessed the strength of the *hlgCB* promoter in the three prototypic strains. A fragment of 434 bp upstream of the *hlgC* start codon (P*hlgC*) from the USA300, ST80, and PEN strains (Fig. S5) was cloned in front of the *gfp* gene in a plasmid that was introduced into the three genetic backgrounds. P*hlgC* of PEN was significantly stronger than P*hlgC* of the two other strains regardless of the genetic background in which the plasmid was expressed. P*hlgC* of USA300 and ST80 had lower but similar strengths in the PEN and ST80 backgrounds, while P*hlgC* of USA300 was slightly stronger than P*hlgC* of ST80 in the USA300 background (Fig. 4A). Strikingly, the observed promoter activities do not perfectly correlate with the steady-state levels of the mRNAs (Fig. 3A) and the levels of proteins (Fig. 1B). As the sequences of the *hlgC* mRNAs differ slightly in the three strains (Fig. 5A), we then studied the impact of the respective *hlgCB* mRNAs on HlgC production independently of the promoter. The *hlgCB* coding sequence from each strain was placed under the control of the P3 promoter (*agr* promoter activated by quorum sensing) and introduced into the three different genetic backgrounds deleted for the *hlg*ABC genes (Fig. 4B). As shown in Fig. 4B, the USA300 sequence, irrespective of the genetic background, produced low levels of HlgC, whereas both the ST80 and PEN sequences allowed efficient HlgC synthesis in all

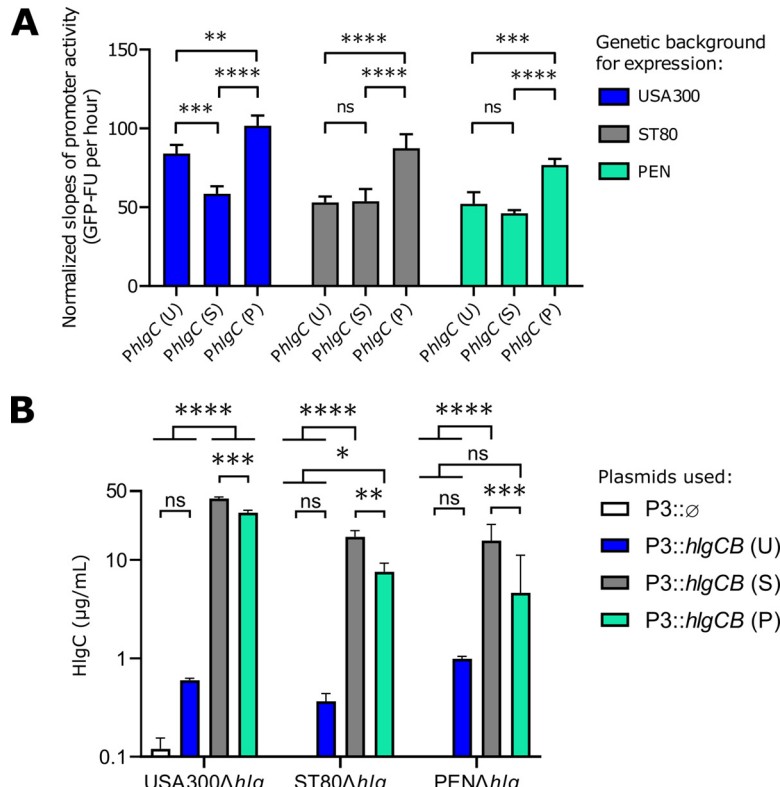

**FIG 4** Differential HlgC production profiles depend on promoters but also posttranscriptional mechanisms. (A) *hlgC* promoter (P*hlgC*) activity was tested according to its original genetic background and when expressed in autologous and heterologous genetic backgrounds. The 434-bp upstream regions, including the *hlgC* ATG codon, from the USA300 [P*hlgC* (U)], ST80 [P*hlgC* (S)], and PEN [P*hlgC* (P)] strains were cloned into the pACL-1484 plasmid in front of the *gfp* gene. Plasmids were transduced into the three strains USA300, ST80, and PEN. Promoter activity was measured by GFP signal quantification over time (GFP fluorescence units [GFP-FU]), during a 24-h kinetic in CCY medium. The promoter activity slopes were estimated between the 4-h and 18-h time points from the means from biological triplicates and normalized to the values for the negative control. (B) Quantification by ELISAs ($n = 3$) of HlgC (in micrograms per milliliter) from biological triplicates of the supernatants from cultures grown for 8 h in CCY medium. The *hlgCB* genes from the USA300 [P3::*hlgCB* (U)], ST80 [P3::*hlgCB* (S)], and PEN [P3::*hlgCB* (P)] strains were cloned into the pCN38 plasmid under the control of the P3 promoter and overexpressed in the USA300 Δ*hlg*, ST80 Δ*hlg*, and PEN Δ*hlg* strains. Plasmid pCN38 without *hlgCB* (P3::ø) was used as a negative control. For the data in panels A and B, Tukey's multiple-comparison test was performed (ns, nonsignificant [adjusted $P$ value {$P_{adj}$} > 0.05]; *, $P_{adj} < 0.01$; **, $P_{adj} < 0.01$; ***, $P_{adj} < 0.001$; ****, $P_{adj} < 0.0001$).

genetic backgrounds, with ST80 *hlgCB* mRNA having slightly better production of HlgC than PEN *hlgCB* mRNA.

Overall, these results demonstrated variabilities in the strength and activity of the *hlgCB* promoter in the different genetic backgrounds. In addition, the few variations in the *hlgC* mRNA sequences (Fig. 5A) contributed to the different levels observed for HlgC independently of the genetic background.

**A single SNP in the 5′ UTR drastically changes *hlgCB* translatability.** We then questioned whether polymorphisms in the nucleotide sequences of the *hlgCB* mRNA may impact translatability, and we first compared the *hlgCB* mRNA sequences of each strain. The USA300 and PEN strains have only a 1-nucleotide (nt) difference (U-G) in their 5′-UTR sequences at position −13 relative to the start codon, close to the ribosome binding site. Of note, ST80 has numerous SNPs in the *hlgC* coding region compared to USA300 and PEN but has the same polymorphism as PEN in the 5′ UTR (Fig. 5A and Fig. S5).

To study the potential impact of the SNPs on translatability, the formation of a simplified initiation complex was analyzed using toeprinting assays. The experiments were performed using *S. aureus* 30S subunits, the initiator tRNA^Met, and *in vitro*-transcribed full-length *hlgCB* mRNAs of the three strains. Using an oligonucleotide specific to *hlgC*, a clear toeprint was

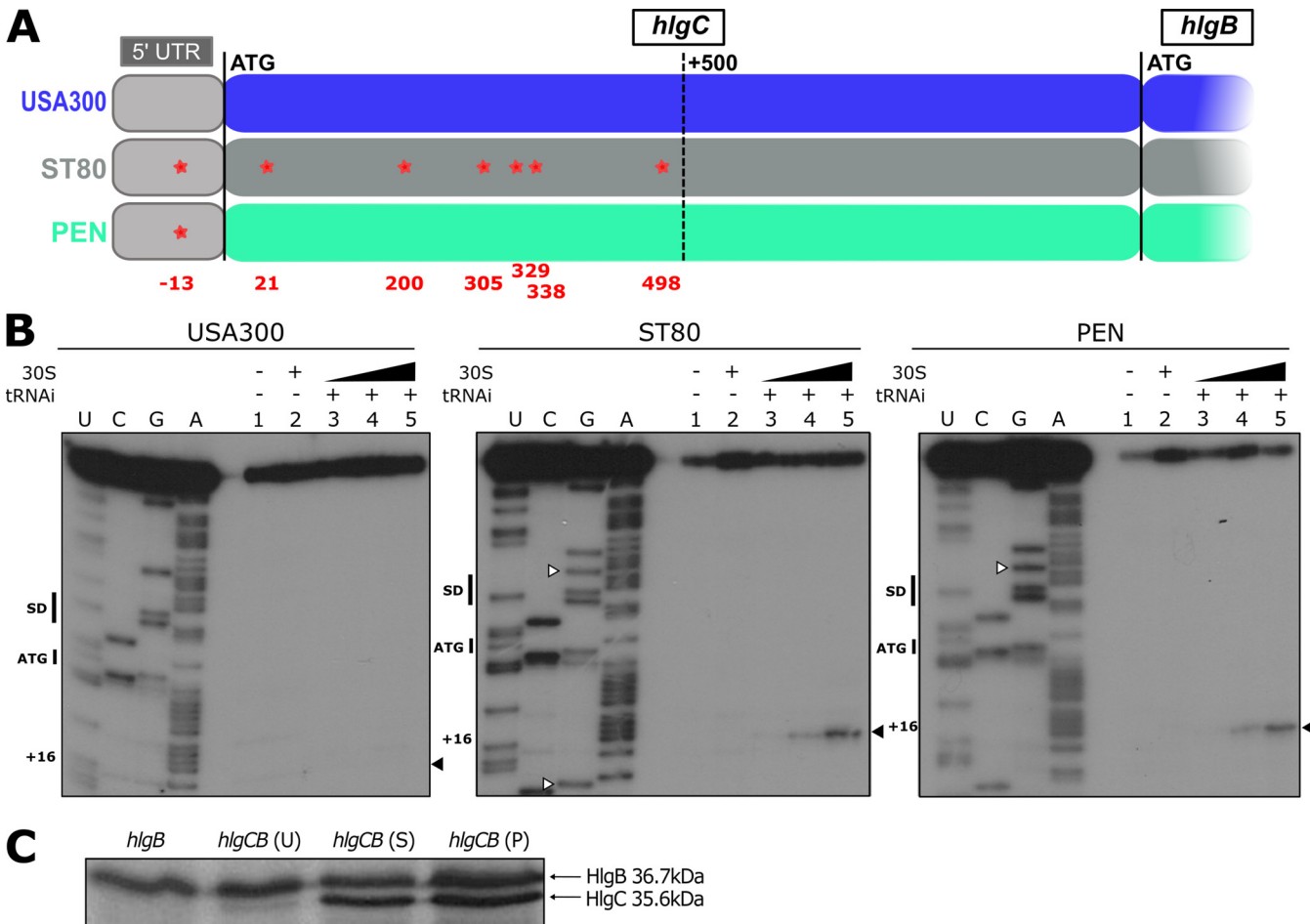

**FIG 5** A single SNP in the 5′ UTR drastically changes *hlgCB* translatability. (A) Schematic representation of the *hlgC* sequence up to *hlg*B with its 5′ UTR (light gray) from the USA300 (blue), ST80 (dark gray), and PEN (green) strains. SNPs between the USA300 and ST80 or PEN strains are shown in red in the schematic representation of the gene, and the nucleotide distance from the ATG codon is shown in red below the gene representation. The PEN and ST80 strains have the same SNP in the 5′ UTR. (B) Toeprint assays showing the binding of *S. aureus* ribosomes on the *hlgCB* transcript in the 5′ UTR of the *hlgC* mRNA, according to the sequence (USA300, ST80, or PEN). Lane 1, incubation control of mRNA alone; lane 2, incubation control of mRNA with 30S subunits; lanes 3 to 5, formation of the ribosomal ternary complex containing mRNA (1 nM), the initiator tRNA$^{fMet}$ (tRNAi) (1 $\mu$M), and increasing concentrations of 30S (lane 3, 0.5 $\mu$M; lane 4, 1 $\mu$M; lane 5, 2 $\mu$M); lanes U, C, G, and A, sequencing ladders. The Shine-Dalgarno (SD) sequence, the translation start site (ATG), SNPs compared to the USA300 sequence (white arrowheads), and the toeprinting signal (+16) are indicated. The experiments were performed at least three times. (C) *In vitro* translation of *hlgB* and *hlgCB* mRNAs using pUC-T7 constructs with *hlgB* or *hlgCB* DNA sequences from the USA300 (U), ST80 (S), and PEN (P) strains. *S. aureus* 70S ribosomes were used, and the reaction was performed with [$^{35}$S]methionine for 4 h. Two independent experiments were performed.

observed at position +16 from the initiation codon of the ST80 and PEN *hlgC* transcripts, but no signal was detected for USA300 mRNA (Fig. 5B). Similar experiments were performed using an oligonucleotide specific to *hlgB* on the three *hlgCB* transcripts, and in comparison, the toeprinting assay was also done with the short *hlgB* mRNA (the sequences of *hlgB* are strictly identical in the three strains). A clear toeprint at position +16 of the *hlgB* ribosome binding site was observed for all transcripts (Fig. S6). These data showed that the *hlgB* ribosome binding site is accessible for the formation of a translation initiation complex, whatever the mRNA context.

Next, *in vitro* translation assays were performed using *S. aureus* 70S ribosomes and *hlgCB* mRNAs of the three strains. The level of the HlgC protein synthesized was much lower from USA300 *hlgCB* mRNA than from the PEN and ST80 transcripts (Fig. 5C). Hence, the HlgB and HlgC levels appeared almost equimolar among ST80 and PEN *hlgCB* mRNAs but not USA300 *hlgCB* mRNA. These results confirm the impact of the 5′-UTR SNP on the translation efficiency of *hlgC*. The experiments also clearly showed that *hlgB* can be translated from both *hlgCB* and *hlgB* mRNAs *in vitro* (Fig. 5C).

Altogether, our data demonstrate that the presence of a single SNP (U-G at position −13

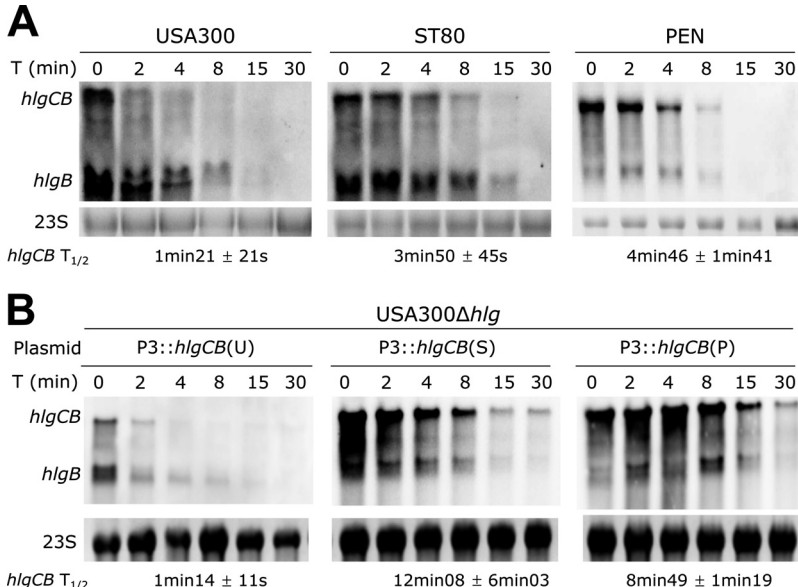

**FIG 6** The absence of *hlgC* translation reduces *hlgCB* mRNA stability. *hlgCB* and *hlgB* mRNA stability upon rifampicin exposure was determined. Cultures of the USA300, ST80, and PEN strains (A) or USA300 Δ*hlg* complemented with the pCN38 plasmid expressing *hlgCB* from USA300 [*hlgCB*(U)], ST80 [*hlgCB*(S)], or PEN [*hlgCB*(P)] under the control of the P3 promoter (B) in CCY medium were treated with rifampicin at 5 h of growth. Total RNA was extracted before rifampicin addition (0-min time point [T]) and 2 min to 30 min after rifampicin addition [T (min)]. *hlgCB* and *hlgB* mRNAs were probed with the *hlgB* probe. Ethidium bromide-stained 23S rRNA (23S) from the same samples was used as a loading control. The calculated half-lives ($T_{1/2}$) of *hlgCB* mRNA relative to 23S are presented below the gel pictures and represent the means from biological triplicates.

relative to the start codon) in the 5′ UTR of *hlgC* has strong consequences for ribosome recruitment and translation efficiency. In addition, the synthesis of HlgB could arise both from the internal entry of the ribosome at the *hlgB* initiation site and from the processed form of *hlgB*.

**mRNA stability is impaired in USA300.** As ribosome binding to mRNA is sufficient to protect mRNA from degradation by RNases in Gram-positive bacteria (18), we thus hypothesized that the *hlgCB* mRNA half-life of USA300 would be reduced compared to those of ST80 and PEN. Therefore, the *hlgCB* and *hlgB* mRNA stabilities were determined for the three strains using rifampicin assays (Fig. 6A). The results suggested that the *hlgCB* mRNA was less stable in the USA300 strain than in the ST80 and PEN strains (1 min 21 s compared to 3 min 50 s and 4 min 46 s, respectively). In contrast, only slight variations were observed for the half-lives of the short *hlgB* transcript among the three strains (Fig. 6A). To avoid the impact of the promoter strength, rifampicin assays were also performed on mRNAs from the USA300 Δ*hlg* mutant strain expressing the different *hlgCB* variants under the control of the *agr*P3 promoter (as shown in Fig. 4B). The data showed that the *hlgCB* mRNA from USA300 was more unstable than the *hlgCB* mRNAs from ST80 and PEN, with a half-life of 1 min 21 s compared to 8 min 6 s and 8 min 46 s, respectively (Fig. 6B).

All in all, our data suggested that the short half-life of the *hlgCB* mRNA in USA300 is correlated with the weak efficiency of *hlgC* translation.

## DISCUSSION

In this study, we first demonstrated that the PVL-negative strain PEN was as virulent as the PVL-positive strains USA300 and ST80 in the rabbit model of pneumonia and that this virulence was driven mainly by HlgCB production. When highly produced, HlgCB leads to rapid mortality, the strong production of cytokines in the lung, and high bacterial loads, similar to what is observed with PVL-positive strains, including community-acquired MRSA (CA-MRSA) strains USA300 and ST80 (16, 17). Consistently, although HlgCB targets the

same receptors as LukS-PV, HlgCB interacts with different motifs, leading to better cell permeabilization (12). Since all *S. aureus* strains possess the *hlg* locus (19, 20), but not all strains cause severe CAP (4), the present study strengthens the hypothesis that the level of expression of this locus matters (15). However, as in the PVL-positive strain, the inactivation of the Hlg locus did not lead to a full attenuation of virulence, one could raise the hypothesis of mutual compensation between the pore-forming toxins.

To decipher the mechanisms leading to the variation in HlgCB synthesis among clinical strains, we investigated various levels of regulation, including promoter activity, translation, and mRNA stability. Differences in promoter activity were expected due to the polymorphisms between the PEN and ST80 (2 SNPs) or USA300 (2 SNPs and 1 insertion) strains (see Fig. S5 in the supplemental material). Variation in expression due to differences in promoter sequences has been reported previously for other toxins such as Hla (21). Surprisingly, the promoter activity of P*hlgC* from USA300 was weaker when expressed in the two other strain backgrounds, suggesting an indirect effect caused by the modulation of an unknown transcriptional regulator. In addition, the large differences in translation efficiencies among the USA300, PEN, and ST80 strains are related to a single SNP in the 5′ UTR of the *hlgCB* operon. Other studies have shown that the introduction of mutations in the 5′ UTRs of mRNAs in *Escherichia coli* and *Bacillus subtilis* can induce changes in the mRNA structure, with consequences for ribosome recruitment at initiation sites and translation efficiencies (22–26). Secondary structure prediction of the *hlgC* 5′ UTR revealed a hairpin structure that partially sequesters the weak Shine-Dalgarno (SD) sequence (Fig. S7). The G-13-U mutation in USA300 could give rise to an additional base pair that would reinforce the stability of the hairpin structure. Interestingly, the three strains that have an HlgCB profile similar to that of USA300 carry either the same SNP at position −13 (FAL003 and MAS222) or an SNP at position −7 (BES288 [BES]). For the BES strain, the SNP at position −7 (G to U) reduces the strength of the SD sequence, and an additional base pair can be formed (Fig. S7). It is thus tempting to propose that these two SNPs (positions −13 and −7) would decrease helix breathing to hinder ribosome binding and/or decrease SD sequence efficiency. In addition, ST80 contains 6 additional SNPs in the coding sequence that do not alter the sequence of HlgC and thus are considered to be silent. However, a recent study performed in the yeast *Saccharomyces cerevisiae* revealed that both synonymous and nonsynonymous mutations in coding sequences resulted in a significant reduction in fitness and that the mutations frequently alter mRNA levels (27). Therefore, we do not exclude that the SNPs in the coding sequence might have consequences on translation speed and/or mRNA stability in ST80. It is well known that ribosome densities on mRNAs have direct consequences on mRNA stabilities in *E. coli* (28), while ribosome binding at the initiation site of mRNAs in *B. subtilis* was sufficient to protect the mRNA against degradation (18). Hence, it is very likely that the stability of the *hlgCB* mRNA depends on its translatability more than on the SNP variations in the coding sequences. Interestingly enough, a very similar mechanism was recently described in *Klebsiella pneumoniae* (29). A single synonymous mutation in the ninth codon of *ompK36* led to the stabilization of a hairpin structure sequestering the SD sequence and in turn repressed the translation of the membrane porin OmpK36. Although this mutation attenuated virulence, it contributed to the emergence of resistance to carbapenems. Here, we show that in USA300, the loss of *hlgC* translation resulted in an mRNA that was more vulnerable to degradation, leading to a low level of production of the virulence factor HlgC.

Considering all of our data, we propose the following scenario to explain the variation in HlgCB levels among clinical strains (Fig. 7). The PEN strain has a strong promoter leading to a large amount of translatable and stable *hlgCB* mRNA, allowing a high level of production of HlgCB. The ST80 strain, despite a weaker *hlgC* promoter than that of PEN, shows better translatability and mRNA stability than USA300 and allows a high level of production of HlgCB. Hence, both the ST80 and PEN strains produce high levels of HlgC and HlgB. In contrast, USA300 has intermediate *hlgC* promoter activity, very poor *hlgC* translation, and an unstable mRNA due to the presence of an SNP in the 5′ UTR of the *hlgC* mRNA, resulting in very low levels of HlgC (Fig. 7). Altogether, these

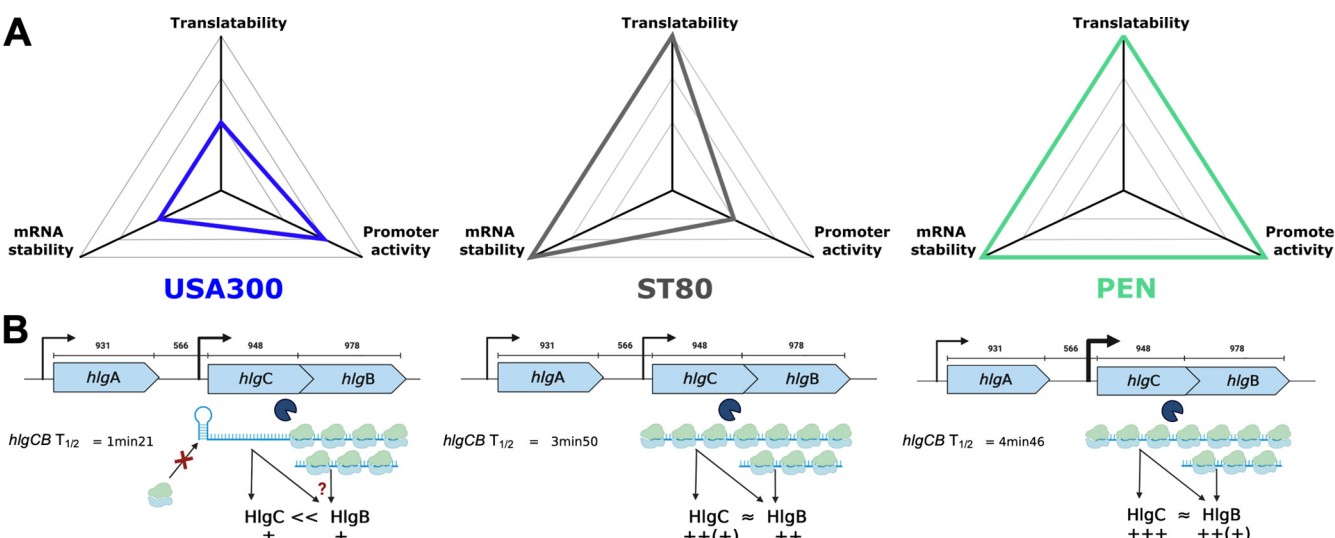

**FIG 7** Regulatory mechanisms impacting gamma-hemolysin production. (A) Kiviat diagram representing the different mechanisms explored in this study and their importance regarding HlgC production in the USA300 (blue), ST80 (gray), and PEN (green) strains, on an arbitrary scale. The larger the area is, the larger the amount of HlgC will be. (B) Model for the different levels of regulation of HlgC and HlgB production. In the USA300 strain, HlgC production is impaired due to an SNP impeding ribosome recruitment at the *hlgC* initiation site. Translational coupling might thus be abolished, and HlgB synthesis might come from the internal entry of the ribosome at the initiation site of *hlgB* on *hlgCB* mRNA or from the processed transcript.

three levels of *hlgC* regulation explain the protein levels measured by absolute quantification in the three strains. Interestingly, although several SNPs among the three strains in and upstream of the *hlgC* sequence were observed and were responsible for the variation in HlgC production, no SNP was identified in *hlgB*, and its protein level was less affected among clinical strains. This might be due to a dual evolutionary constraint as HlgB also forms heterodimers with HlgA. The high level of variability in coregenome gene expression raises the question of the evolutionary pressure that led to the selection of this expression diversity. It may represent a fine and specific adaptation to the context, such as interactions with the host in which each strain develops. It is very likely that other determinants modulate the overall level of HlgCB production in clinical strains, such as transcriptional or posttranscriptional regulators. As illustrated in this study and elsewhere, epistasis effects as well as the noncoding and coding sequences all play roles in the expression of virulence factors (30–32). The continuous increase in the number of genome-wide association studies (GWASs) (31, 33, 34) performed for the analysis of large sets of strains may allow the identification of these other contributors in the future.

The presence of a new *hlgB* transcript adds another layer of complexity. Our data strongly suggested that this new transcript is not a bona fide mRNA but resulted from the maturation/degradation of *hlgCB* induced by as-yet-unknown RNases, which could include RNase III, RNase J1/J2, RNase Y, or others. Previous RNA sequencing performed on different strains showed the presence of an antisense RNA against *hlgC* and more *hlgB* than *hlgC* transcripts (35), supporting the existence of a processed form of *hlgB*. Whether the antisense RNA is involved in the maturation process of *hlgB* remains to be determined. The processed *hlgB* mRNA appears to be translated *in vitro*, and in the case of USA300 where *hlgC* translation is impeded, leading to an unstable mRNA, the synthesis of HlgB might essentially originate from the *hlgB* mRNA. As the HlgB component is used for both the HlgCB and HlgAB toxins (36), the *hlgB* transcripts could be translated under specific conditions to allow the production of HlgB independently of HlgC in order to form the HlgAB toxin. Unlike *hlgC* and *hlgB*, *hlgA* is located upstream of the *hlgCB* operon and possesses its own promoter (37, 38). On the one hand, *hlgCB* is upregulated by the *agr* system (39, 40), SarA (39, 40), and the SaeRS regulation system (41–45) but is downregulated by *rot* (46); on the other hand, *hlgA* is regulated mostly by the SaeRS system (43, 44, 47, 48). The dissociated regulation of *hlgA* and

*hlgB* remains unexpected since HlgA requires HlgB to induce pore formation (49). One hypothesis is that the *hlgB* transcript might be appropriate under specific conditions to optimize the formation of the HlgAB toxin. To further complicate the matter, nonequimolar levels of production of HlgA and HlgB may be relevant under certain conditions as HlgA alone antagonizes its receptors (CXCR1, CXCR2, and CCR2), preventing neutrophil activation (11), while HlgB alone induces the autocrine motility factor receptor (AMFR) pathway, leading to inflammation (50).

This work illustrates the complexity of virulence factor expression in clinical strains, where a given gene can produce a protein at highly variable levels, not only due to the corresponding promoter activity. Furthermore, proteins from an operon may not be expressed equimolarly, and proteins that belong to different operons but are supposed to form an equimolar assembly may not be produced at equivalent levels. It is very likely that such phenomena will be discovered in other species as they probably participate in the evolutionary plasticity of bacteria.

## MATERIALS AND METHODS

**Bacterial strains and growth conditions.** All strains and plasmids used in this study are listed in Table S2 in the supplemental material. *S. aureus* strains were routinely cultured overnight in tryptic soy broth (TSB; Difco), diluted to an optical density at 600 nm ($OD_{600}$) of 0.05 in CCY medium (3% [wt/vol] yeast extract, 2% Bacto Casamino Acids, 2.3% sodium pyruvate, 0.63% disodium phosphate, and 0.041% potassium phosphate), and grown for 5 h (rifampicin test) or 8 h at 37°C with shaking. When required, chloramphenicol was added at a concentration of 10 $\mu$g/mL.

For animal experiments, strains were inoculated into 5 mL of CCY medium and incubated overnight at 37°C with shaking. Next, 500 $\mu$L of this culture was inoculated into CCY medium and agitated for 4 h at 37°C to reach the exponential growth phase.

**Construction of mutant strains.** The *S. aureus hlgC* stop codon mutants were created in the clinical PEN strain and the ST80 strain using the pnCasSA-BEC genome-editing system as previously described (51) (see the supplemental material). The entire *hlg*ACB locus was also deleted in the USA300, ST80, and PEN strains by a double-recombination process as previously described (52) (see the supplemental material).

**Absolute quantification of HlgC and HlgB in clinical strains using targeted proteomics.** *S. aureus* clinical strains were cultured for 8 h in CCY medium and centrifuged at 10,000 rpm for 5 min at 4°C to separate the bacterial pellets. For each strain, 50 $\mu$L of the supernatant (50 $\mu$g of proteins) was reduced, alkylated, and digested using a LysC-trypsin mix (Promega) (53). To quantify HlgC and HlgB, the following two signature peptides were selected and purchased in an isotopically labeled version (AQua QuantPro; Thermo Fisher Scientific) to serve as quantification standards: YVSLINYLP[$^{13}C_6$,$^{15}N_2$]K (HlgC) and SNFNPEFLSVLSH[$^{13}C_6$,$^{15}N_4$]R (HlgB). These labeled peptides were added to the digested supernatant at a final concentration of 2.1 pmol/mL. Next, the digest was desalted on a $C_{18}$ ZipTip (MacroSpin Harvard) before drying by vacuum centrifugation and storage at $-20$°C. The dried peptide digest was solubilized in 40 $\mu$L of a solution containing 2% acetonitrile and 0.1% formic acid, and 6 $\mu$L of this solution was analyzed via targeted proteomics. Targeted proteomic analyses were performed on a 6500 QTrap mass spectrometer (AB Sciex) operating in the selected reaction mode (SRM). Liquid chromatography (LC) separation was performed on an Ultimate 3000 system (Thermo Scientific). The parameters for LC separation and scheduled SRM acquisitions are presented in the supplemental material. LC-SRM data analysis was performed using Skyline software. All transitions were individually inspected and excluded if they were deemed unsuitable for quantification (low signal-to-noise ratio or obvious interference). Unlabeled/labeled peak area ratios were calculated for each SRM transition, and these ratios were used to determine the corresponding mean peptide ratios. The HlgC and HlgB concentrations in the *S. aureus* supernatants were calculated based on the mean ratio of each signature peptide.

**Rabbit model of pneumonia.** Immunocompetent New Zealand White male rabbits (body weight of 2.8 to 3.2 kg) were challenged at Vivexia (Dijon, France) by the intrabronchial instillation of 0.5 mL of *S. aureus* strains diluted to 9.49 to 9.61 $\log_{10}$ CFU/mL in a sterile saline solution (0.9% NaCl) (Otec) to induce pneumonia. The growth of all strains was monitored by $OD_{600}$ measurements after culturing for 8 h in CCY medium, and no difference was observed (Table S3). The inoculum was gently flushed through the tracheal catheter under laryngoscopic control. The catheter was immediately removed, and the animals were rapidly extubated and allowed to go back to their cage. The postinfection survival rate of the rabbits was then recorded 2 to 3 times a day. If they had not spontaneously died, the animals were euthanized 48 h after challenge. The lungs were collected for cytokine and bacterial load estimations: each lung (right and left) was weighed and homogenized in 10 mL of a sterile saline solution. Bacterial densities in crude lung homogenates were determined by plating 10-fold dilutions onto Chapman agar and incubating the plates for 24 h at 37°C. Bacterial concentrations were expressed as $\log_{10}$ CFU per gram. All procedures in the protocol were approved by the local ethics committee for animal experiments and by the Ministère de l'Enseignement Supérieur et de la Recherche (APAFIS approval number 22710-201911071356264 v1).

**Cytokine quantification.** The cytokines IL-8 and IL-1$\beta$ were quantified by enzyme-linked immunosorbent assays (ELISAs) using a rabbit IL-1$\beta$ ELISA kit and a rabbit IL-8 ELISA kit (Invitrogen). Aliquots of

2 mL of the lung homogenate shredded in a saline solution were centrifuged for 30 min at 10,000 rpm at 4°C, before appropriate dilution in an assay diluent solution. ELISAs were then performed strictly according to the supplier's protocols. The final concentration was reported based on the total mass of the lung. Three biological replicates were used for each strain tested.

**Northern blot analysis.** Total RNA was prepared from 10 mL of a culture grown for 8 h according to the manufacturer's procedures for the FastRNA pro blue kit (MP Biomedicals) with the FastPrep apparatus (MP Biomedicals). Electrophoresis of total RNA (10 to 30 $\mu$g) was performed on a 1.5% agarose gel containing 25 mM guanidium thiocyanate. After migration, RNAs were transferred by a capillary onto positively charged nylon membranes (GE Healthcare Life Sciences). Hybridizations with specific digoxigenin (DIG)-labeled probes complementary to *hlgA*, *hlgC*, *hlgB*, Rsal, and 5S rRNA sequences (Table S4) followed by luminescence detection were carried out as previously described (54).

**Primer extension assays.** To determine the transcriptional start sites of *hlgCB* and *hlgB* mRNAs, primer extension assays were performed as previously described (55). Briefly, 30 $\mu$g of total RNAs from the USA300 and ST80 strains was reversed transcribed with avian myeloblastosis virus (AMV) reverse transcriptase (New England BioLabs [NEB]) using 5′-radiolabeled primers (PEC1 for *hlgC* and PEB1, PEB2, and PEB3 for *hlgB*) (Table S4). The sequencing ladders were obtained from *hlgCB* or *hlgB* PCR products (amplified with the primers listed in Table S4) and the Vent (Exo⁻) DNA polymerase (NEB). All samples were fractionated on a 10% polyacrylamide–8 M urea gel in 1× Tris-borate-EDTA (TBE). After migration, the gel was exposed to autoradiography film.

**Promoter activity assays.** The regions upstream of *hlgC* (434 bp) and *hlgB* (929 bp), including their start codons, were amplified from USA300, PEN, and ST80 genomic DNAs (gDNAs) for *hlgC* and only ST80 gDNA for *hlgB* (because no sequence differences were observed among the three strains) using oligonucleotides P*hlgC*-EcoRI-F and P*hlgC*-XbaI-R and oligonucleotides P*hlgB*-PuvII-F and P*hlgB*-KpnI-R for the *hlgC* and *hlgB* upstream regions, respectively (Table S4). Next, the amplicons were fused to the *gfpuv* gene by ligation into the pACL1484 plasmid (56) using the appropriate restriction enzymes (Table S4). The 16S rRNA promoter was cloned into pACL1484 as a positive control, as previously described (57). All constructs were verified by PCR and sequencing using oligonucleotides pACL-F and pACL-R (Table S4). Plasmids were first electroporated into the RN4220 strain before transformation by electroporation into the USA300, ST80, and PEN strains. A 96-well plate (catalog number 655090; Greiner Bio-One) was inoculated with 200 $\mu$L per well of the culture at an OD$_{600}$ of 0.05 in CCY medium containing 10 $\mu$g/mL of chloramphenicol and incubated at 37°C for 24 h. GFP measurements were performed with an Infinite 200Pro apparatus (Tecan), with an excitation wavelength of 488 nm and an emission wavelength of 535 nm. Both the GFP signal and the OD$_{600}$ were measured every 15 min for 24 h. Each strain was evaluated in triplicate.

**Overexpression of *hlgCB* mRNA.** The sequence from nt −39 upstream of the *hlgC* ATG codon to nt +60 after the *hlgB* stop codon was amplified with oligonucleotides *hlgCB*-PstI-F and *hlgCB*-BamHI-R (Table S4) from gDNAs of the USA300 [*hlgCB* (U)], ST80 [*hlgCB* (S)], and PEN [*hlgCB* (P)] strains. Next, the amplicons were cloned into the pCN38-P3 plasmid using the appropriate restriction enzymes downstream of the P3 promoter. All constructs were verified by PCR and sequencing. Plasmids were first electroporated into RN4220 before transformation by electroporation into the USA300 Δ*hlg*, ST80 Δ*hlg*, and PEN Δ*hlg* strains. All strains were cultivated overnight in TSB with chloramphenicol, subcultured to an OD$_{600}$ of 0.05 in CCY medium without antibiotics, and incubated for 8 h. The supernatants were centrifuged for 5 min at 10,000 rpm at 4°C and stored at −80°C.

**HlgC quantification by an ELISA.** HlgC quantification was performed by a sandwich ELISA using a custom-made mouse monoclonal antibody (R&D Biotech) and a custom-made polyclonal rabbit F(ab)′$_2$ biotinylated antibody (R&D Biotech), both raised against HlgC. A 96-well Nunc MaxiSorp plate (Thermo Scientific) was coated with anti-HlgC monoclonal antibody at 10 $\mu$g/mL overnight at 20°C. After 5 consecutive washes with phosphate-buffered saline (PBS)–0.05% Tween (PBS-T), wells were saturated for 1 h 30 min at 20°C with a blocking solution containing PBS-T, low-fat milk (5 g/L), and bovine serum albumin (BSA) (1 g/L). Standard dilutions (15 to 1,000 ng/mL) of recombinant HlgC or the culture supernatant were denatured for 1 h at 95°C, loaded in duplicate, and incubated for 2 h at 37°C. The plate was washed, polyclonal rabbit F(ab)′$_2$ biotinylated antibody (1.55 $\mu$g/mL) was added, and the plate was incubated for 1 h 30 s at 37°C. After the incubation, the plate was washed, and ExtrAvidin-peroxidase antibody (Sigma) targeting the biotin molecule and conjugated with horseradish peroxidase (HRP) was added, and the plate was incubated for 1 h at 20°C. A final wash was performed before 75 $\mu$L of the substrate tetramethylbenzidine (KPL SureBlue; SeraCare) were added. The reaction was stopped with sulfuric acid at 1 N. The plates were read at 450 nm in a Bio-Rad model 680 microplate reader.

**Toeprinting assays.** *hlgCB* and *hlgB* mRNA transcripts were synthesized from linearized pUT7 vectors (58). The *hlgCB* sequences (2,007 nt) from the USA300, ST80, and PEN strains and the *hlgB* sequence (1,156 nt) from the ST80 strain, containing their 5′ UTRs, were amplified by PCR using primers pUCT7-*hlgC*-StuI-F or pUCT7-*hlgB*-StuI-F and pUCT7-*hlgB*-BamHI-R (Table S4) containing the T7 promoter. For *hlgB* mRNA, the chosen 5′ UTR was the largest one, including 136 nt upstream of the initiation codon (Fig. 3C). After linearization by BamHI and *in vitro* transcription using T7 RNA polymerase, the DNA template was digested with DNase I (Sigma), and the RNAs were purified using phenol-chloroform-isoamyl alcohol, precipitated in absolute ethanol, and cleaned up with a Micro Bio-Spin Chromato column (Bio-Rad) (54). Toeprinting assays were performed on *hlgCB* and *hlgB* transcripts. The RNAs were first annealed to the appropriate 5′-end-radiolabeled oligonucleotide as previously described (54). A ternary ribosomal complex containing purified *S. aureus* 30S ribosomes (59) (0.5, 1, and 2 $\mu$M), the initiator tRNA$^{fMet}$ (1 $\mu$M), and RNA transcripts (1 nM) was formed at 37°C. After reverse transcription, phenol-chloroform extraction was performed, followed by nucleic acid precipitation. The labeled DNA fragments were then sized by electrophoresis on an 8% polyacrylamide–urea gel. Sequencing ladders were run in parallel to identify the toeprint signals.

***In vitro* translation assays.** The translatability of *hlgCB* and *hlgB* mRNAs was assessed using the PURExpress Δ ribosome kit (NEB) supplemented with purified *S. aureus* 70S ribosomes (59). The ribosomes were first reactivated by incubation at 37°C for 10 min. The reactions were performed for 4 h at 37°C with various plasmids (200 ng of pUT7::*hlgCB* for the USA300, ST80, and PEN strains and pUT7::*hlgB* for the ST80 strain) and *S. aureus* 70S (30 pmol) in a solution containing 5 $\mu$L of buffer A, 1.5 $\mu$L of factors, 1 $\mu$L of RNasin (Promega), and 1 $\mu$L of [$^{35}$S]methionine. Laemmli buffer was added to the samples, which were loaded onto an SDS–15% PAGE gel, with the Page Ruler Plus protein product (Thermo Fisher) run in parallel. After migration, the gel was heat-vacuum transferred onto Whatman paper and exposed to autoradiography film.

**RNA stability assays.** *hlgCB* and *hlgB* mRNA stabilities were determined by rifampicin assays. Briefly, the USA300, ST80, and PEN strains were grown for 5 h in 100 mL of CCY medium, USA300 Δ*hlg* strains complemented with pCN34-P3::*hlgCB* plasmids were grown for 5 h in 100 mL of CCY medium with chloramphenicol, and all cultures were then treated with 1 mL of rifampicin at 30 mg/mL to inhibit transcription. Total RNA was extracted before (time zero) and after rifampicin addition at 2, 4, 8, 15, and 30 min. Northern blot analyses were performed using 23S rRNA as a loading control.

**Data availability.** The genomes of the clinical strains used in this study are available in the ENA repository under accession number PRJEB54685. The PEN HlgC<Q63X mutant genome is available in the ENA repository under accession number PRJEB61327.

## SUPPLEMENTAL MATERIAL

Supplemental material is available online only.

**SUPPLEMENTAL FILE 1**, DOCX file, 3.6 MB.

## ACKNOWLEDGMENTS

We thank Sylvère Bastien for bioinformatics support; Xanthe Adams for technical help; Christiane Wolz, Peter Redder, and Paul Verhoven for providing strains; Iuilia Macavei and Jérôme Lemoine for proteomics analysis; and Véréna Landel from the Hospices Civils de Lyon (France) for help in manuscript preparation.

This work was funded by the ANR (grant ANR-16-CE11-0007-01, Ribostaph) and the RHU IDBIORIV.

We declare no conflicts of interest.

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
