## [Reviewer comments · Microbiology Spectrum]

Microbiology Spectrum

Complex regulation of gamma-hemolysin expression impacts *Staphylococcus aureus* virulence

Mariane Pivard, Isabelle Calderlari, Virgine Brun, Delphine Croisier, Michel Jaquinod, Nelson Anzala, Benoit Gilquin, Chloé Teixeira, Yvonne Benito, Florence Couzon, Pascale Romby, Karen Moreau, and François Vandenesch

Corresponding Author(s): François Vandenesch, Centre International de Recherche en Infectiologie

Review Timeline:

Submission Date:	March 15, 2023
Editorial Decision:	April 3, 2023
Revision Received:	May 22, 2023
Accepted:	May 25, 2023

Editor: M.-N. Frances Yap

Reviewer(s): The reviewers have opted to remain anonymous.

Transaction Report:

DOI: <https://doi.org/10.1128/spectrum.01073-23>

April 3, 2023

Prof. François Vandenesch
Centre International de Recherche en Infectiologie
rue G. Paradin
Lyon 69008
France

Re: Spectrum01073-23 (Complex regulation of gamma-hemolysin expression impacts *Staphylococcus aureus* virulence)

Dear Prof. Vandenesch:

Your manuscript has been reviewed by three experts in the field. While the reviews are overall positive, a few important points are raised that should be addressed prior to further consideration. In particular, quantitative and statistical analyses should be performed in all experiments involving multiple biological replicates. Additionally, please provide justification if the requested genetic complementation can not be fulfilled.

Link Not Available

Sincerely,

M.-N. Frances Yap

Journals Department
Reviewer comments:

Reviewer #1 (Comments for the Author):

In the manuscript entitled "Complex regulation of gamma-hemolysin expression impacts *Staphylococcus aureus* virulence" the authors describe their observations on the structure and expression of the hlgCB operon from several pathogenic strains of *S. aureus*. The authors analyzed the expression of the operon's proteins and observed high variability among strains. They continue their analysis using two strains that expressed the panton-Valentine leucocidin (PVL) system and one strain that does not express such pore-forming toxin but is still pathogenic. The manuscript correlates the expression of the hlgC gene with pathogenicity on the strain lacking of the PVL system.

Their analysis concluded that each strain expressed differently the hlgCB operon, whereas transcription, translation and mRNA stability played a role in changing the expression mostly of the hlgC gene among the strains. A unique mechanism is proposed for one of the strains, where a single nucleotide change at the 5'-UTR affects translation initiation of the hlgC, perhaps affecting the stability of a secondary structure that could sequester the Shine-Dalgarno.

The experiments shown in this manuscript are clean and clearly show that the studied strains expressed differently their hlgCB operon. A major criticism could be that the authors did not show if the pathogenesis of PVL positive bacteria is affected in the absence of expression of the hlgC gene as shown for the PVL minus strain (the authors have produced strains without hlgCB, Figure 4). Such results could give the authors more indications about the interaction between both PVL and hlgA-hlgCB systems. Also, the discussion does not address this PVL and hlgA-hlgCB possible interaction as well.

Minor concerns

- 1) Have the authors detected the hlgA expression of all strains shown at Fig 1? It seems based on Fig3A that it could be also different among strains. If so, perhaps it could be useful to show that data as well.
- 2) Figure 2 and 4, perhaps adding the actual P values would be more accurate for the reviewer to understand the statistical significance of these observations.
- 3) Figure 3A, second panel. This reviewer wondered if the authors did not detect a bigger mRNA containing hlgA and hlgC.
- 4) Figure 3B, would the promoter for the 16S rRNA gene be good control (too strong)? Why not use the promoter found in front of the hlgC gene?
- 5) Figure 5A, the authors should consider showing the actual DNA sequence of the regions of importance. This would help the reader understand their findings.
- 6) Page 3, line 39 change "promotor" by "promoter"
- 7) Page 8, line 141 -142, Consider change the sentence by " The results show three 5'-ends located at -80, -104, and -136 upstream the hlgB start codon and one at -35 upstream the hlgC start codon, all possible transcriptional start sites"
- 8) Page 8, line 163. "As shown in Figure 1" Is that correct? Or should it be Figure 4B?
- 9) Page 14, line 307 Change "Material" by "Materials"
- 10) The section of Materials and Methods. Consider using the names instead of the molecular formulas for all used compounds.
- 11) Figure S3, Would not the RNase J1/J2 act on the possible independent hlgB transcript as well?
- 12) Figure S5B, what is the size of the upstream region from the start codon of the hlgB transcript? It seems odd that the 5' end of the primer extension is similar to that observed with the whole hlgCB transcript (Fig SBA). Also, lane 1 does not show any extension.

Reviewer #2 (Comments for the Author):

The manuscript by Pivard et al examined the regulation mechanisms of HlgCB in Staphylococcal clinical isolates. A series of experiments revealed complex regulation mechanisms at both transcriptional and translational levels. Interestingly, a SNP in 5'UTR was found to impair hlgC translation. Overall, the manuscript is of high quality: it provides detailed and new insights into virulence gene regulation in *S. aureus*; experiments were well designed and carefully done; conclusions are well-supported and the manuscript is very well written. I only have one suggestion and some minor points.

1. Suggestion: there is no complementation in Figure 2. It would be more convincing if the PEN 2. Figure 1. Proteins found in the supernatant are not total proteins. It does not necessarily correlate with mRNA and protein production in the cell. Please modify the corresponding text.
3. Line 157: I think the promoter activities do correlate with mRNA level in Fig. 3A
4. Line 163: should be "As shown in Figure 4B"
5. Line 174: "position -13", does it mean -13 relative to start codon? Please be clearer. Also, indicate what SNP it is.
6. Line 194: "presence of a single SNP", again, directly indicate the SNP nt here.
7. Figure 3 legend: it is confusing with "All identified +1 are in red", would be clearer to replace "+1" with mRNA start sites.
8. Figure 5A: indicate the SNP nt in the figure
9. Is there a tendency that PVL negative strains produce higher levels of HlgCB, and vice-versa? It would be interesting to discuss this.

Reviewer #3 (Comments for the Author):

This paper investigates the involvement of the bi-component pore-forming leucocidin HlgCB in the ability of *S. aureus* to cause pneumonia. HlgCB is a core-genome encoded virulence factor, however its abundance can vary significantly between different clinical isolates. Indeed, despite HlgC and HlgB being encoded in the same operon, levels of HlgC can differ significantly to HlgB. In this paper, the authors firstly show the variation in expression of HlgC in a range of different isolates. Three isolates, representing distinct HlgCB expression profiles and PVL status, all displayed virulence in an animal model of acute pneumonia, with virulence in PVL-negative strains dependent on the production of HlgC. To determine the mechanism behind this variability in expression of HlgC between the different isolates, the authors use a variety of techniques and unveil that HlgC cell levels are determined by the promoter strength of the hlgCB operon, on mRNA stability, and on translation efficiency.

This is a well written paper describing some robustly performed experiments. Below are some suggestions for improvement.

Minor corrections

- All figure legends should state how many times an experiment was performed, especially if representative images are used eg Fig. 3A.
- For all blots, please use densitometry to quantify changes in expression level.
- Fig 2A - the authors should include a growth curve for this 4 strains to show that the virulence defect for PEN Q63X is not due to a replication defect.
- Fig 4B - statistic should be included here, as for 4A.
- Fig 7 - how were the T1/2 figures determined? These are not mentioned in the text.
- Fig S3 show that the 5' end of the hlgCB mRNA is also processed. This should be mentioned in the main text and not just referred to in the figure legend.

Grammatical

- Line 52: Please check the writing of hlgCB throughout, as sometimes the entire gene is in italics and sometimes not. eg line 52 with hlgC and hlgCB.
- Line 101: Instead of saying "the first group" please refer to them as the isolates from profile 2 for clarity.
- Line 138 and elsewhere: pALC1484 plasmid misspelt as pACL.
- Fig. S3 legend - here the authors use a mix of a comma and prime symbol for 5'. Prime should be used.

Staff Comments:

Preparing Revision Guidelines

Please return the manuscript within 60 days; if you cannot complete the modification within this time period, please contact me. If you do not wish to modify the manuscript and prefer to submit it to another journal, please notify me of your decision immediately so that the manuscript may be formally withdrawn from consideration by Microbiology Spectrum.

Corresponding authors may join or renew ASM membership to obtain discounts on publication fees. Need to upgrade your

membership level? Please contact Customer Service at Service@asmusa.org.

Reviewer #1 (Comments for the Author):

In the manuscript entitled "Complex regulation of gamma-hemolysin expression impacts *Staphylococcus aureus* virulence" the authors describe their observations on the structure and expression of the *hlgCB* operon from several pathogenic strains of *S. aureus*. The authors analyzed the expression of the operon's proteins and observed high variability among strains. They continue their analysis using two strains that expressed the panton-Valentine leucocidin (PVL) system and one strain that does not express such pore-forming toxin but is still pathogenic. The manuscript correlates the expression of the *hlgC* gene with pathogenicity on the strain lacking of the PVL system.

Their analysis concluded that each strain expressed differently the *hlgCB* operon, whereas transcription, translation and mRNA stability played a role in changing the expression mostly of the *hlgC* gene among the strains. A unique mechanism is proposed for one of the strains, where a single nucleotide change at the 5'-UTR affects translation initiation of the *hlgC*, perhaps affecting the stability of a secondary structure that could sequester the Shine-Dalgarno.

The experiments shown in this manuscript are clean and clearly show that the studied strains expressed differently their *hlgCB* operon. A major criticism could be that the authors did not show if the pathogenesis of PVL positive bacteria is affected in the absence of expression of the *hlgC* gene as shown for the PVL minus strain (the authors have produced strains without *hlgCB*, Figure 4). Such results could give the authors more indications about the interaction between both PVL and *hlgA-hlgCB* systems. Also, the discussion does not address this PVL and *hlgA-hlgCB* possible interaction as well.

Answer: The role of *HlgCB* in the pathogenesis of PVL-positive strains is of interest although not exactly in the scope of the present study that focused on the role of *HlgCB* in PVL negative strains. However, we performed the rabbit model with individual mutants of *HlgC*, PVL, *Hla* as well as a triple mutant in the ST80 genetic background. The triple mutant was the only one to completely attenuate the virulence of ST80, suggesting a compensation of the different toxins between them. It is possible that in animals infected with the single mutants, the in-situ expression of the other toxins is increased but we have not been able to obtain reliable and reproducible assays of the different toxins from rabbit infected lungs; we therefore cannot conclude whether this hypothesis is true or not. We have added the results of the above additional animal experiments in the result section and as a supplementary figure, and discussed the hypothesis of the probable compensation of the different toxins between them.

Minor concerns

1) Have the authors detected the *hlgA* expression of all strains shown at Fig 1? It seems based on Fig3A that it could be also different among strains. If so, perhaps it could be useful to show that data as well.

Answer: As mentioned in the discussion of the manuscript, *HlgA* synthesis is under the control of its own promoter which is stimulated by the presence of blood (Malachowa N, et al. PLOS ONE. 2011;6:e18617) and is not correlated to *HlgCB* synthesis. We have tested all strains shown in Figure 1 for *HlgA*. However, as the culture medium used (CCY media) was not

appropriate for optimal expression of HlgA, the values were all in a very low range (mostly 0 to 2, only three strains with more than 5 pmole/mL) (see Figure below). Since this information does not add much to the understanding, we propose not to include these results in the manuscript.

2) Figure 2 and 4, perhaps adding the actual P values would be more accurate for the reviewer to understand the statistical significance of these observations.

Answer: Statistics have been added on Figures 2D and 4B.

3) Figure 3A, second panel. This reviewer wondered if the authors did not detect a bigger mRNA containing hlgA and hlgC.

Answer: the RNAs detected by the hlgC and hlgB probes had a similar migration on the agarose gel and their sizes were very similar when compared with molecular weight markers. At this specific position of the gel, no signal was detected with the hlgA probe. Furthermore, using the *hlgA* probe we have detected a single mRNA with a size compatible to *hlgA* mRNA alone.

4) Figure 3B, would the promoter for the 16S rRNA gene be good control (too strong)? Why not use the promoter found in front of the *hlgC* gene?

Answer: We have constructed a *PhlgC::gfp* control which also gave a positive signal compared to the result obtained with the putative *PhlgB* construct. We have added this figure in the supplement material section.

5) Figure 5A, the authors should consider showing the actual DNA sequence of the regions of importance. This would help the reader understand their findings.

Answer: we have followed the reviewer suggestion and added the sequence comparison of hlgCB in the three strains, in the supplementary figure S5.

6) Page 3, line 39 change "promotor" by "promoter"

Answer: corrected

7)Page 8, line 141 -142, Consider change the sentence by " The results show three 5'-ends located at -80, -104, and -136 upstream the *hlgB* start codon and one at -35 upstream the *hlgC* start codon, all possible transcriptional start sites"

Answer: We have included this change

8)Page 8, line 163. "As shown in Figure 1" Is that correct? Or should it be Figure 4B?

Answer: We are thankful to the reviewer, and we have corrected this mistake.

9)Page 14, line 307 Change "Material" by "Materials"

Answer: corrected

10)The section of Materials and Methods. Consider using the names instead of the molecular formulas for all used compounds.

Answer: We have modified the text accordingly

11)Figure S3, Would not the RNase J1/J2 act on the possible independent *hlgB* transcript as well?

Answer: we agree with the reviewer that RNases J1/J2 can be involved in the maturation process of *hlgB*. Nevertheless, we also do not exclude that other RNases such as RNase III or RNase Y, might also participate in this process. We have added this point in the discussion section.

12)Figure S5B, what is the size of the upstream region from the start codon of the *hlgB* transcript? It seems odd that the 5' end of the primer extension is similar to that observed with the whole *hlgCB* transcript (Fig S5A). Also, lane 1 does not show any extension.

Answer: These experiments were performed with the large *hlgCB* mRNA (2007 nucleotides, 643 kDa) and the processed *hlgB* mRNA (1156 nucleotides, 371 kDa), which were both *in vitro* transcribed. The 5' untranslated region of the *hlgB* transcript contains 136 nucleotides upstream the start codon corresponding to the largest 5' end detected with the primer extension experiments. To reveal the positioning of the ribosomal subunit on *hlgB* ribosome binding site, we used a 5' end-labeled oligonucleotide complementary to the coding sequence of *hlgB* mRNA, 132 nucleotides downstream the start codon. The experiments performed on *hlgCB* and *hlgB* mRNAs were not performed in parallel, and the samples did not migrate on the same gel. Nevertheless to avoid any ambiguities and based on the reviewer' comments, we have performed again the toe-printing assays on the *hlgB* mRNA where we have previously controlled the size of the mRNA on a denaturing agarose gel electrophoresis. The data showed a typical toeprint at position +16 corresponding to the 3' edge of the ribosome binding site in the presence of the initiator tRNA. The sequence of *hlgB* was annotated on the left side of the autoradiography showing the expected size of the RNA. In lane 4, the amount of material is higher than in the other lanes. As the radiolabelled primer was added at the beginning of the assays, we cannot exclude that the hybridization was more efficient in this assay or alternatively that we have lost some material during the precipitation in the other lanes. However, this experiment showed that *in vitro* the ribosome is able to significantly bind to the mRNA. We have added a comment in the legend of the figure S5 (now S6). In the revised

manuscript, we have also added more details in the Material and methods concerning the size of the RNA transcripts.

We just would like to stress that the two transcripts (*hlgCB* and *hlgB*) were also used in parallel for *in vitro* translation assays. The data were well correlated with the Toe-printing assays and clearly showed that the short *hlgB* transcript produced equivalent HlgB protein yields than the corresponding long *hlgCB* transcripts. Taken together, these data showed that the ribosome binding at *hlgB* is almost equivalent in the two transcripts and that the translation of *hlgB* occurred independently of *hlgC*.

Reviewer #2 (Comments for the Author):

The manuscript by Pivard et al examined the regulation mechanisms of HlgCB in Staphylococcal clinical isolates. A series of experiments revealed complex regulation mechanisms at both transcriptional and translational levels. Interestingly, a SNP in 5'UTR was found to impair *hlgC* translation. Overall, the manuscript is of high quality: it provides detailed and new insights into virulence gene regulation in *S. aureus*; experiments were well designed and carefully done; conclusions are well-supported and the manuscript is very well written. I only have one suggestion and some minor points.

1. Suggestion: there is no complementation in Figure 2. It would be more convincing if the PEN<Q63X can be complemented by HlgC. If it is difficult to do rabbit infection experiments, the authors should soften the statement in the text.

Answer: We agree that a complementation experiment is always welcome. We did not perform it because the sequencing revealed only one SNP difference between the wild type and the PEN<Q63X strain in a non-coding region (Genome access on ENA with the accession number PRJEB61327, now cited in the supplemental methods). This observation made a new round of animal experimentation unjustifiable in view of the ethical considerations on animal experimentation..

2. Figure 1. Proteins found in the supernatant are not total proteins. It does not necessarily correlate with mRNA and protein production in the cell. Please modify the corresponding text.

Answer: we have amended the sentence accordingly

3. Line 157: I think the promoter activities do correlate with mRNA level in Fig. 3A

Answer: we partly agree as the promoter of the PEN strain appears the strongest and the *hlgCB* signal on the Northern blot is also the strongest. Conversely, the promoter of the ST80 strain is equivalent or inferior to the promoter of USA300, however the *hlgCB* signal on the Northern blot is stronger in ST80 as compared to USA300. We have thus amended the sentence to temper the assertion that "there was no correlation" to "there was not a perfect correlation".

4. Line 163: should be "As shown in Figure 4B"

Answer: we have corrected the sentence

5. Line 174: "position -13", does it mean -13 relative to start codon? Please be clearer. Also, indicate what SNP it is.

Answer: We have added the missing information

6. Line 194: "presence of a single SNP", again, directly indicate the SNP nt here.

Answer: we have added this information in the revised version.

7. Figure 3 legend: it is confusing with "All identified +1 are in red", would be clearer to replace "+1" with mRNA start sites.

Answer: We have introduced the modifications as suggested by the reviewer

8. Figure 5A: indicate the SNP nt in the figure

Answer: we have followed the suggestion of the reviewer.

9. Is there a tendency that PVL negative strains produce higher levels of HlgCB, and vice-versa? It would be interesting to discuss this.

Answer: we have previously performed a semi-quantitative proteomic analysis of exotoxins on 136 strains isolated from severe CAP (doi.org/10.3389/fcimb.2023.1162617). In this paper we could see a weak negative correlation between Lus-PV and HlgC (figure 4 of the above cited paper) but it did not reach significance perhaps owing to the sample size. We have included this point in the introduction.

Reviewer #3 (Comments for the Author):

This paper investigates the involvement of the bi-component pore-forming leucocidin HlgCB in the ability of *S. aureus* to cause pneumonia. HlgCB is a core-genome encoded virulence factor, however its abundance can vary significantly between different clinical isolates. Indeed, despite HlgC and HlgB being encoded in the same operon, levels of HlgC can differ significantly to HlgB. In this paper, the authors firstly show the variation in expression of HlgC in a range of different isolates. Three isolates, representing distinct HlgCB expression profiles and PVL status, all displayed virulence in an animal model of acute pneumonia, with virulence in PVL-negative strains dependent on the production of HlgC. To determine the mechanism behind this variability in expression of HlgC between the different isolates, the authors use a variety of techniques and unveil that HlgC cell levels are determined by the promoter strength of the hlgCB operon, on mRNA stability, and on translation efficiency.

This is a well written paper describing some robustly performed experiments. Below are some suggestions for improvement.

Minor corrections

- All figure legends should state how many times an experiment was performed, especially if representative images are used eg Fig. 3A.

Answer: We have added this information in all the figures. Usually, we are performing at least three independent replicates.

- For all blots, please use densitometry to quantify changes in expression level.

Answer: We have quantified the gels (shown in figure 6) and added the half-lives below the autoradiographies.

- Fig 2A - the authors should include a growth curve for these 4 strains to show that the virulence defect for PEN Q63X is not due to a replication defect.

Answer: We did not follow the growth but measured the OD at 8h culture from which the inoculum was prepared. The four strains had very similar OD. We added these data in a new supplementary table.

- Fig 4B - statistic should be included here, as for 4A.

Answer: As mentioned for the reviewer 1, statistics have been added in figures 2D and 4B

- Fig 7 - how were the T1/2 figures determined? These are not mentioned in the text.

Answer: Half-lives of the transcripts have been added in the figure legend and the method for their determination was also added.

- Fig S3 show that the 5' end of the hlgCB mRNA is also processed. This should be mentioned in the main text and not just referred to in the figure legend.

Answer: We have followed the reviewer suggestion and have mentioned this point in the main text.

Grammatical

- Line 52: Please check the writing of hlgCB throughout, as sometimes the entire gene is in italics and sometimes not. eg line 52 with hlgC and hlgCB.

Answer: according to the reviewer' suggestion, we have checked and corrected these mistakes all along the manuscript.

- Line 101: Instead of saying "the first group" please refer to them as the isolates from profile 2 for clarity.

Answer: as suggested by the reviewer, we have modified the sentence.

- Line 138 and elsewhere: pALC1484 plasmid misspelt as pACL.

- Fig. S3 legend - here the authors use a mix of a comma and prime symbol for 5'. Prime should be used.

Answer: these mistakes have been corrected

May 25, 2023

Prof. François Vandenesch
Centre International de Recherche en Infectiologie
rue G. Paradin
Lyon 69008
France

Re: Spectrum01073-23R1 (Complex regulation of gamma-hemolysin expression impacts *Staphylococcus aureus* virulence)

Dear Prof. Vandenesch:

Your manuscript has been accepted, and I am forwarding it to the ASM Journals Department for publication. You will be notified when your proofs are ready to be viewed.

Sincerely,

M.-N. Frances Yap
Editor, Microbiology Spectrum
